

# A systematic review on diabetic retinopathy detection and classification based on deep learning techniques using fundus images

Dasari Bhulakshmi and  Dharmendra Singh Rajput

School of Computer Science Engineering and Information Systems, Vellore Institute of Technology, Vellore, Tamil Nadu, India

## ABSTRACT

Diabetic retinopathy (DR) is the leading cause of visual impairment globally. It occurs due to long-term diabetes with fluctuating blood glucose levels. It has become a significant concern for people in the working age group as it can lead to vision loss in the future. Manual examination of fundus images is time-consuming and requires much effort and expertise to determine the severity of the retinopathy. To diagnose and evaluate the disease, deep learning-based technologies have been used, which analyze blood vessels, microaneurysms, exudates, macula, optic discs, and hemorrhages also used for initial detection and grading of DR. This study examines the fundamentals of diabetes, its prevalence, complications, and treatment strategies that use artificial intelligence methods such as machine learning (ML), deep learning (DL), and federated learning (FL). The research covers future studies, performance assessments, biomarkers, screening methods, and current datasets. Various neural network designs, including recurrent neural networks (RNNs), generative adversarial networks (GANs), and applications of ML, DL, and FL in the processing of fundus images, such as convolutional neural networks (CNNs) and their variations, are thoroughly examined. The potential research methods, such as developing DL models and incorporating heterogeneous data sources, are also outlined. Finally, the challenges and future directions of this research are discussed.

## INTRODUCTION

Diabetes mellitus (DM) is a dangerous condition that is typically brought on by excessive blood sugar levels. Among the organs that are frequently negatively impacted by diabetes over an extended period are the kidneys, brain, and optic nerves. Diabetes affects people of all ages and is a significant public health issue (*Mayya, Kamath & Kulkarni, 2021*; *Tsiknakis et al., 2021*; *Tripathi & Kumar, 2020*). In Fig. 1, we discuss the diabetes status, present and future, through this diabetes prediction chart. According to the International Diabetes Federation (IDF), in the year 2021, approximately 537 million individuals (20–79 years old) were affected by diabetes. By 2030, there will be 643 million diabetics worldwide, and

Corresponding author
Dharmendra Singh Rajput,
dharmendrasingh@vit.ac.in

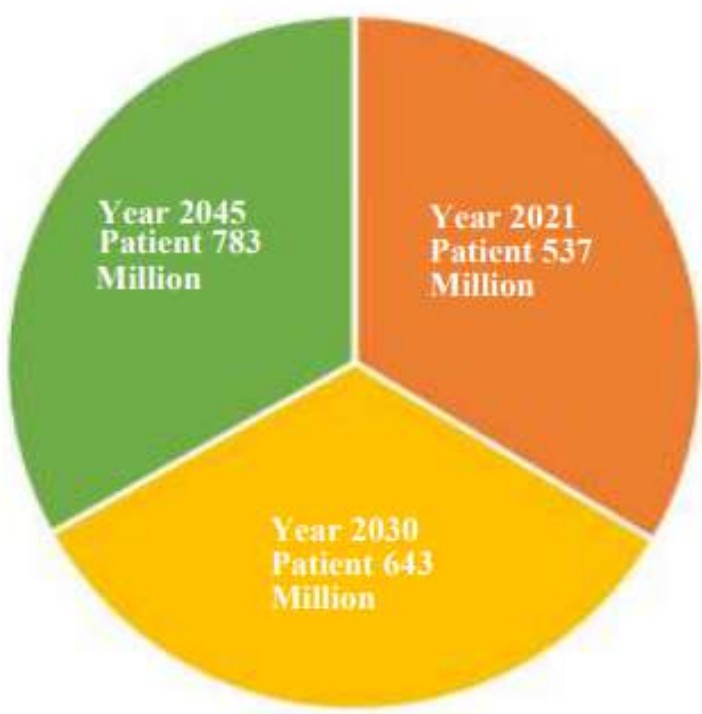

**Figure 1** **Diabetes prediction chart.**

by 2045, there will be 783 million. Adults with diabetes suffer for three out of every four people worldwide (*Fregoso-Aparicio et al., 2021*; *Rajput et al., 2022*). Type 1, type 2, and gestational diabetes are three types of diabetes. Type 1 requires insulin and is common in young children. Type 2 is more prevalent in older and obese patients due to insulin resistance. Gestational diabetes, which occurs during pregnancy, affects glucose cell use and can harm both the mother and the unborn child. It is primarily caused by hyperglycemia (*Aslan & Sabanci, 2023*; *Sivaranjani et al., 2021*).

DR is a condition that can lead to blindness in working-age people due to microvascular complications of diabetes. This condition can be prevented, and it is characterized by hemorrhages, microaneurysms, and changes in the retinal microvasculature. There are various treatment options available for retinal conditions, including anti-vascular endothelial growth factor (anti-VEGF) injections, vitrectomy, laser photocoagulation, neuroprotective drugs, and stem cell therapy for retinal regeneration *Salamat, Missen & Rashid (2019)*. Before the advent of ML techniques, medical professionals predominantly relied on manual fundus image assessment and traditional methods for diagnosing DR. Several non-ML techniques are still employed for DR detection. Doctors utilize specialized cameras to capture high-resolution color photographs of the retina, revealing blood vessels, optic nerves, and other retinal components. Fluorescent dye injections highlight blood vessels, aiding in identifying leaky spots and abnormal blood vessel development (*Barros et al., 2020*). Fundus photography provides detailed information, while optical coherence tomography (OCT) offers cross-sectional images, aiding in determining retinal layers and

macular edema. Visual acuity testing helps assess the impact of diabetic retinopathy on eyesight. The international clinical DR severity scale categorizes severity based on fundus images. Despite these informative techniques, the process can be time-consuming, and early diagnosis may be challenging. Incorporating artificial intelligence and ML approaches has significantly enhanced the efficiency and accuracy of DR detection in recent years (*Kalyani et al., 2023*).

To differentiate between a healthy retina and one impacted by DR, medical professionals, and ML techniques analyze various features and anomalies in retina fundus images. They typically focus on crucial signals and characteristics, including microaneurysms, hemorrhages, exudates, cotton wool spots, neovascularization, macular edema, optic nerve head abnormalities, vessel tortuosity and beading, general vascular changes, and background retinopathy grading (*Thanki, 2023*). These features serve as common inputs for ML techniques and training models capable of autonomously recognizing and categorizing DR. The algorithms are trained to identify patterns and anomalies akin to those a human specialist would observe while manually examining fundus photos.

DL is a part of the ML technique that learns from examples and filters information like the human brain. It is used in everyday tasks and is the core technology for driverless cars, such as detecting stop signs and guiding pedestrians (*Sarker, 2021*). The relationship between DL, ML, and artificial intelligence (AI) can be seen in Fig. 2. DL, a versatile ML technique, utilizes deep neural networks (DNNs), CNNs, RNNs, and other techniques to process data in various application sectors.

DL models play a crucial role in autonomously learning intricate hierarchical representations from data, especially for detecting DR in fundus images. Traditional image analysis techniques often struggle with capturing complex patterns in fundus pictures, highlighting the necessity of DL models, particularly deep CNNs, for accurate DR diagnosis (*Ahmed et al., 2023*). DL models can extract meaningful information from raw pixel values, comprehending abstract details as they traverse through layers. End-to-end training enables efficient classification of different disease phases or identification of healthy fundus pictures. DL thrives with large labeled datasets, and the availability of annotated fundus photos enhances their generalizability. DL models are well-suited for early intervention, facilitating quick DR recognition, and can be continuously improved to adapt to new information, enhancing performance over time (*KELLIL, 2023*). The objective of our study is to investigate the most recent advancements in automated scientific methods for detecting DR and classification, including conventional and DL approaches. Particularly, this review indicates a few suggestions and future requirements for increasing the significant effectiveness of DL models in detecting DR.

## Contributions of the article

The contributions of our work can be summarized as follows.

1. First, we present a survey methodology and related works & research motivation of the concepts of DR and DR detection methods in ML and DL based on retinal features.
2. Second, we describe in detail the DR and databases that are used for DR detection and classification.

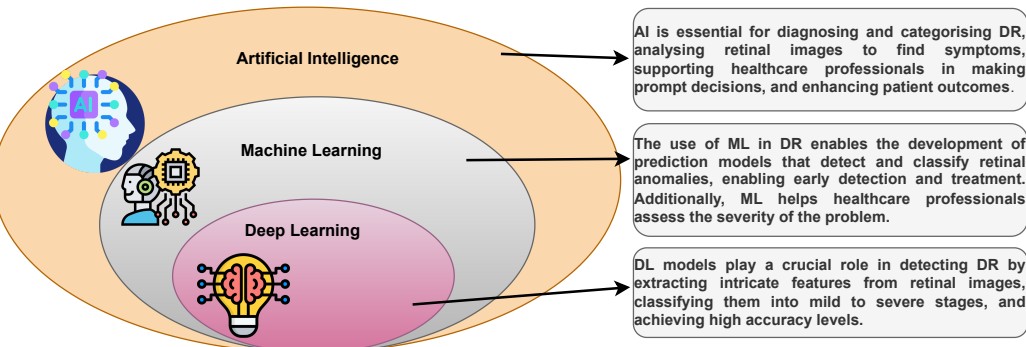

**Figure 2  The family of DL with DR.**

3.  Third, we describe in detail ML, DL, and FL in DR detection and classification.
4.  Fourth, we describe in detail the applications of ML, DL, and FL and the role of each technology in DR.
5.  Fifth, we provide details of the metrics for measuring performance used to evaluate DR detection methods.
6.  Despite several research and development activities, the associated challenges and issues in the implementation of DR in DL are identified. Finally, the potential future research directions towards detecting DR in DL are highlighted.

The review article is organized as follows: 'Survey methodology' describes the survey methodology. 'Related works & research motivation' describes related works and research motivation carried out by various researchers on classification techniques for detecting DR. 'Diabetic retinopathy DR' covers various technologies and publicly available datasets. 'Machine learning in diabetic retinopathy' discusses the ML in the DR. 'Deep learning in diabetic retinopathy' uses DL in DR, and 'Federated learning in diabetic retinopathy' uses FL in DR. 'Applications of ML, DL, and FL' discusses applications of ML, DL, and FL. 'Evaluation performance metrics' discusses the evaluation performance metrics, and 'Challenges and future directions' discusses challenges and future directions. 'Conclusion' concluded with the contribution of the work and remarks.

## SURVEY METHODOLOGY

In this review, a literature survey is chosen to provide an overview of the role of DR detection and classification based on DL techniques using fundus images in DL, which includes the following steps: Initially, we highlight the limitations of existing survey articles in the fields of ML and DL techniques. The next step is to search for relevant scientific and research articles on the detection of DR. We focus on peer-reviewed high-quality articles in relevant and reputed journals, conferences, symposiums, workshops, and books. The articles were found through searches of electronic databases well-known research repositories include IEEE, Science Direct, Google Scholar, PubMed, Scopus, Springer, and Elsevier. The titles and abstracts of the articles are used for screening, and the eligibility criteria are used to assess the authors' contributions to the articles. Ensuring that the full-text

articles are eligible. The studies that were included in the systematic and qualitative review. The inclusion of the research articles completely examined the subject of DR eye disease prediction; the research articles have clear core objectives; the articles use the ML and DL methods to detect eye diseases are added. The only acceptable research articles are those written in English. The excluded research articles are duplicate publications; studies that were not fully published in the full text; letters, comments, case studies, and seminar reports; any articles that use ML and DL but are unrelated to detecting eye diseases; studies that were not published in English.

## RELATED WORKS & RESEARCH MOTIVATION

DR is a diabetes complication impacting the blood vessels in the retina. Untreated, it leads to vision impairment and eventual blindness. Regular eye exams enable early detection when the condition is more manageable, preserving vision. Effective management, including medication, lifestyle changes, and medical treatments, can decrease the risk of blindness or severe visual loss. Vision loss significantly impacts the quality of life.

*Gupta et al. (2023)* suggest a collaborative learning approach using multiple datasets, preserving privacy, and using public images for testing and private datasets for training. Transfer learning is used to retrain the CNN with the lowest performance. *Hassan et al. (2023)* enhanced optical coherence tomography (EOCT) model has been developed to improve retinal optical coherence tomography (OCT) classification performance using modified ResNet and random forest techniques. The model's accuracy, sensitivity, specificity, precision, false discovery rate, and Matthew's correlation coefficient were measured. *Sengar et al. (2023)* diagnostic prediction is made by an eye-deep network and a multi-layer neural network for extracting the key features from fundus images for performing baseline models and utilizing statistical methods. This section includes a comprehensive discussion of DR detection methods (*Mali & Jadhav, 2021*).

Table 1 shows more about the current state of DR research and a summary of important DL implementations of different DR datasets.

The motivation for research into the detection of DR and its potential to enhance healthcare, avoid vision loss, increase accessibility to healthcare, reduce costs, and progress in the fields of DL and computer vision is the driving force behind research into the identification of DR using DL. This research has a significant impact that helps individual patients as well as the healthcare system.

## DIABETIC RETINOPATHY

Individuals with prediabetes, characterized by elevated blood pressure, cholesterol, and blood sugar, are prone to DR. Fluctuations in blood sugar levels can damage the delicate blood vessels in the eyes, potentially leading to complete blockages. This forces the eye to create unexpected, tiny blood vessels, but occasionally these blood vessels are too small and may leak fluid into the retina, causing various lesions to appear on the retinal image.

**Table 1  Summary of important DL implementations of different DR datasets.**

| Ref No | Dataset | Methodology | Evaluation metrics | Objectives achieved | Drawbacks |
|---|---|---|---|---|---|
| *Butt et al. (2022)* | APTOS dataset | Pre-trained CNN models | Accuracy, Precision, Recall, F-measure | Various techniques were used to remove noise and artifacts from the uploaded images | Lack of comparison and explanation of the other methods and classifiers. |
| *Parthasharathi et al. (2022)* | Retinal dataset | CNN model | Accuracy, Sensitivity, Precision, Specificity, and F1-Score | The VGG19 has been successfully used to develop a CNN model that could detect DR | The author does not provide a clear comparison of the proposed hybrid model with other existing methods for DR detection. |
| *Dayana & Emmanuel (2022)* | DIARETD0 | optimized DNN with Chronological Tunicate Swarm Algorithm (CTSA) | F1-Score, Sensitivity, Specificity, and Accuracy | Screening for additional eye-related disorders such as diabetic macular edema and glaucoma | A new optimization algorithm with improved features can enhance classification efficiency for maximum performance. |
| *Pan et al. (2023)* | Fundus images | Inception V3 and ResNet-50 | Accuracy, Precision, Recall and F1-score | Adoption of CAD will set an end to incorrect diagnoses caused by blurry images, a lack of personal expertise, and other factors | The author does not compare its results with other existing methods for fundus image classification. |
| *Jiwani, Gupta & Afreen (2022)* | IDRiD dataset | DL technique | AUC, Sensitivity, Accuracy and Specificity | CNN model is applied for DR classification. The system is trained using the ADAM optimizer | Lack of explanation on the choice of parameters and architecture. |
| *Chavan & Choubey (2023)* | Messidor-1, Messidor-2, and Kaggle | Transfer learning is allowed. EfficientNet B3 and Fine Tuning enabled ResNet 101, TL-EN3, and FT-RN 101 | Accuracy, Sensitivity, Specificity and F1 score | Future studies will examine other eye-related conditions such as diabetic macular edema and glaucoma | Lack of details on the feature extraction and fusion. |
| *Özbay (2023)* | EyePacs | ADL-CNN | Accuracy, Sensitivity, Specificity, and F-measure | It is possible to employ DR detection with various CNN parameters in many multimedia applications | Failed to explain the use of artificial bee colony algorithm for image segmentation and its advantages over other methods. |
| *Ratna et al. (2023)* | APTOS 2019 Blindness Detection dataset | CNN and ResNet architecture | Accuracy, Precision, Recall | CNNs and ResNet were especially effective in enhancing the precision of automated DR diagnosis using fundus pictures | There is insufficient explanation for the architecture and hyperparameter selections. |

## Types of lesions

- Microaneurysms (Mas), small saccular dilations of capillaries, are the earliest clinical signs of DR, causing vision loss due to endothelial barrier local loss of functions.
- Haemorrhages (HM) are large blood accumulations in the retina, larger than 125 micrometers, beginning in the superficial layers of the retina.
- Hard exudates (EX) are yellow spots on the retina caused by lipoprotein leaks from plasma, with sharp margins and possibly located in the outermost layers of the retina.
- Soft exudates (SE), also known as cotton wool, form on the retina when an arteriole is occluded due to nerve fiber expansion.
- Glaucoma occurs when blood vessels form in the cornea, increasing pressure inside the retina and causing damage to the optic nerve, which transmits images to the brain's cortex (*Kumar et al., 2020*).

DR is classified into different stages based on the severity of retinal changes, including no DR, mild NPDR, moderate NPDR, severe NPDR, and proliferative DR. In Fig. 3 shown all the stages of DR.

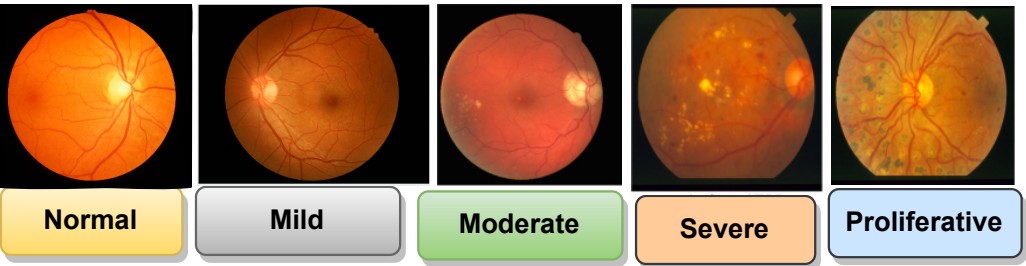

**Figure 3** **Stages of DR (*Qummar et al., 2019*).** Images source credit: Prasanna Porwal, Samiksha Pachade, Ravi Kamble, Manesh Kokare, Girish Deshmukh, Vivek Sahasrabuddhe, Fabrice Meriaudeau, April 24, 2018, ''Indian Diabetic Retinopathy Image Dataset (IDRiD)'', IEEE Dataport, https://dx.doi.org/10.21227/H25W98. CC BY 4.0 DEED.

## Databases

Detection of DR and vessels can be achieved using public and private data sets, including retinal imaging methods like OCTs and fundus images. OCT scans reveal retinal thickness and contour details, while fundus images are 2-dimensional (*Vij & Arora, 2023*). The Table 2 includes details about the sample size, resolution, accessibility, data sources, and a list of fundus retinal data sets relevant to DR detection.

## MACHINE LEARNING IN DIABETIC RETINOPATHY

Diabetes is on the rise, leading to a growing number of individuals experiencing DR due to prolonged high blood sugar, resulting in retinal damage. The author introduced a computer vision-based method for analyzing and predicting diabetes from retinal input images. This aids in the early-stage identification of DR by employing support vector machine (SVM) and CNN algorithms, which are then compared (*Ali & Dawood, 2022*).

Diabetes-related retinopathy is a progressive disease causing macular edema, especially in working-age populations in undeveloped countries. Early identification is crucial for preserving eyesight. Diagnosing DR is time-consuming and resource-intensive. ML and DL algorithms have helped detect DR early. This study divides the DR condition into binary classification and multi-classification categories (*Kumar et al., 2020*; *Atwany, Sahyoun & Yaqub, 2022*). Diabetes-related DR is a vision issue that can cause permanent damage if not treated promptly. This study proposes an automated method using CNN and fuzzy classifiers to identify and categorize DR phases using the STARE, DIARETDB0, and DIARETDB1 databases (*Geetha Ramani, Balasubramanian & Jacob, 2012*). The DL technique detects DR, a complex disease caused by high glucose levels that affect the optical nerve. The model, trained on 35126 retinal images, achieved an accuracy of 81% (*Chaudhary & Ramya, 2020*; *Thorat et al., 2021*) Automatic detection systems for DR are being developed to speed up and reduce costs. However, the accuracy of these systems is unsatisfactory due to the scarcity of reliable data. A collaborative learning strategy has been proposed, with the highest accuracy rates among widely used techniques. This early

**Table 2  Summary of the public datasets that are accessible.**

| Dataset | Sample size | Resolution | Ref | Source link |
|---|---|---|---|---|
| Kaggle | 88,702 Fundus images | Different image resolution | *Ghnemat (2022)* | https://www.kaggle.com/c/diabetic-retinopathy-detection |
| E-ophtha | 463 Fundus images | Different image resolution | *Chen et al. (2021)* | https://www.adcis.net/en/thirdparty/e-ophtha/ |
| DIARET DB0 Standard diabetic retinopathy database calibration level 0 | 130 fundus images | 1,500 × 1,152 pixels | *Bilal et al. (2022)* | https://www.kaggle.com/search?q=diaretdb0 |
| DIARET DB1 Standard diabetic retinopathy database calibration level 1 | 89 Fundus images | 1,500 × 1,152 pixels | *Reguant, Brunak & Saha (2021)* | https://www.kaggle.com/datasets/nguyenhung1903/diaretdb1-standard-diabetic-retinopathy-database |
| STARE (Structured Analysis of the Retina) | 400 Fundus images | 700 × 605 pixels | *Saranya, Pranati & Patro (2023)* | https://www.kaggle.com/datasets/vidheeshnacode/stare-dataset |
| DRIVE (Digital retinal images for vessel extraction) | 40 Fundus images | 768 × 584 pixels | *Wang et al. (2015)* | https://drive.grand-challenge.org/ |
| Messidor | 1200 Fundus images | 2,304 × 1,536 pixels | *Bhardwaj, Jain & Sood (2021)* | https://www.adcis.net/en/third-party/messidor/ |
| Messidor-2 | 1,748 Fundus images | Diferent image resolution | *Bilal et al. (2022)* | https://www.adcis.net/en/third-party/messidor2/ |
| FAZ (foveal avascular zone) | 60 Fundus images | 320 × 320 pixels | *Abbas et al. (2017)* | https://www.openicpsr.org/openicpsr/project/117543/version/V2/view?path=/openicpsr/117543/fcr:versions/V2/Diabetic.zip&type=file |
| DR1 | 5776 Fundus images | 640 × 480 pixels | *Li et al. (2017)* | https://www.kaggle.com/datasets/drskprabhakar/dr1-dr2-dr3-image-split-dataset |
| DR2 | 920 Fundus images | 867 × 575 pixels | *Naqvi, Zafar & ul Haq (2015)* | https://www.kaggle.com/datasets/drskprabhakar/dr1-dr2-dr3-image-split-dataset |
| ROC (Retinopathy online challenge) | 100 Fundus images | 768 × 576 to 1,389 ×1,383 pixels | *Mateen et al. (2020)* | http://webeye.ophth.uiowa.edu/ROC/ |
| HRF (High-resolution fundus) | 45 Fundus images | 3,504 × 2,336 pixels | *Zhao et al. (2018)* | https://www.kaggle.com/datasets/avanya456/hrf-images-for-dr |
| DDR | 13,673 Fundus images | Different image resolution | *Gu et al. (2023)* | https://github.com/nkicsl/DDR-dataset |

detection helps prevent visual loss (*Odeh, Alkasassbeh & Alauthman, 2021*; *Das, Biswas & Bandyopadhyay, 2022*).

# DEEP LEARNING IN DIABETIC RETINOPATHY

The field of DL in DR is showing huge potential. By utilizing DL methods, namely CNNs, experts have created models that can analyze retinal images to detect signs of DR. These models assist in early screening and diagnosis by classifying images according to disease stages. CNNs had been trained to recognize anomalies and patterns diagnostic of DR using massive datasets of retinal images. Their accuracy in identifying this disease has been outstanding, often outperforming human specialists. This study uses AlexNet and Resnet101 feature extraction techniques to automatically recognize and categorize DR fundus pictures based on severity, allowing patients with high blood sugar-related DR to avoid complications and potentially lose their eyesight (*Fayyaz et al., 2023*; *Uppamma & Bhattacharya, 2023*). Early diagnosis through automatic detection utilizing modern technology can help prevent consequences like eyesight loss. DL-based approaches are now heavily preferred for creating DR detectors, mainly due to the advancement of AI technologies, in general, *Sebastian et al. (2023)*; *Mishra et al. (2022)*. This section discusses major DL concepts related to diabetic DR. It highlights the importance of converting RGB images to perfect grayscale and resizing them using a DL approach. This process is transmitted to a CNN, simplifying the classification of diabetic and healthy retinal images (*Pak et al., 2020*). It was decided to make an automatic DR scoring system that could recognize pictures based on the pathology of the disease at four different levels of severity. A CNN merges an input picture with a weight matrix to pull out certain image traits while keeping information about how the image is arranged in space (*Soni & Rai, 2021*).

## Types of deep learning

Different kinds of DL techniques are used to diagnose DR by analyzing retinal images. Here are some common types of deep learning techniques.

### *Convolutional neural networks*

The ability of CNNs to automatically learn and extract relevant data from images provides an effective tool for identifying DR from retinal images. Here is how CNNs are used in this situation. The use of CNNs in the diagnosis of DR, a disease that affects diabetics' vision, is considered a landmark. CNNs are especially well-suited for this purpose since they are excellent at image detection and analysis. CNNs are used to analyze retinal images to improve image quality. Convolutional layers, one type of layer in these networks, scan the image being processed using filters to find features like edges, textures, and patterns. These features are then separated and categorized to correspond with various retinopathy severity levels (*Uysal & Güraksin, 2021*). After training, the CNN uses backpropagation and optimization methods to reduce discrepancies between the anticipated output and the actual labels. Cross-validation techniques are used to evaluate the trained CNN on distinct validation and test datasets, ensuring robustness and preventing overfitting (*Sotoudeh-Paima et al., 2022*). DR is a disorder that affects the eyes. CNNs have improved

the diagnosis of this disease. They are efficient in detecting and analyzing images, detecting hemorrhages, microaneurysms, and other signs in retina scans. CNNs can recognize complex patterns, specific changes in blood vessels, lesions, or anomalies and categorize images into various phases. They can help ophthalmologists by doing preliminary analysis, which can assist in detecting missed abnormalities or assess a diagnosis provided by a human specialist (*Islam et al., 2020*).

### Recurrent neural networks

In the context of DR, RNNs have shown promise in enhancing diagnostic accuracy for DR, a leading cause of vision impairment and blindness. RNNs, designed for sequential data, have been utilized in healthcare for various purposes, including the detection and treatment of DR. The RNNs are useful for identifying trends and changes across multiple visits and tracking the disease's direction in retinal images. They can automatically extract pertinent elements from retinal images while learning and adjusting to the particular traits of each patient (*Shilpa & Karthik, 2023*). By examining previous retinal scans and clinical data, RNNs can forecast the risk that diabetic retinopathy will advance to a more severe state. They can help diagnose early, detect small changes, and customize therapies. Additionally, RNNs can accelerate the analysis and classification of retinal images, reducing the strain on medical personnel and facilitating quicker diagnosis (*Mercaldo et al., 2023*). RNNs are useful instruments for treating and avoiding DR overall. RNNs are applied in the context of DR. To collect temporal dependencies and changes in retinal images, RNNs are crucial in medical imaging activities. They can help with accurate diagnosis and observing the evolution of the illness by noticing small variations or anomalies associated with DR. Additionally, RNNs aid in early identification, prevent serious vision damage, and enable personalized treatment plans (*Selvachandran et al., 2023*).

### Generative adversarial networks

Since most of their use is to produce the latest information, such as photographs, rather than categorize or diagnose medical disorders, GANs are not often used directly to diagnose DR. However, there are many ways that GANs can indirectly help with DR diagnosis and treatment. GANs are an effective method for creating artificial retinal images with known features, such as the various phases of DR. These artificial images can add to a small dataset by giving machine learning models, such as CNNs used for diagnosis, more training instances (*Luo et al., 2022*). In addition to creating accurate retinal images, replicating various phases of DR, and exploring disease progression or therapy response, GANs can help researchers. They may improve the clarity of retinal images, facilitating medical specialists' identification of anomalies linked to DR. As they can learn typical retinas and identify deviations as anomalies, GANs may also be used to detect abnormalities in retinal images (*Bellemo et al., 2019*). This aids in the creation and assessment of fresh diagnostic techniques.GANs are used in the context of DR. By creating synthetic retinal images, balancing class distribution, detecting stages, and maintaining patient data, GANs improve DR models. Additionally, they develop improved, enhanced versions, modify models for various other fields, and improve the current understanding of how diseases progress (*Agarwal & Bhat, 2023*).

# FEDERATED LEARNING IN DIABETIC RETINOPATHY

The decentralized ML technique, FL, involves multiple collaborators working together to build a common model without disclosing their raw data. Based on privacy considerations surrounding medical data, it can be especially helpful in the situation of DR. With the use of FL, a decentralized ML technique, multiple collaborators can train a common model without disclosing their raw data. In cases of DR where patient data is sensitive, this is extremely beneficial. Through the use of FL, healthcare organizations may jointly build a strong model without exchanging patient data. Only model updates are sent and combined for the global model; every single organization trains the model using its local data. This method helps to comply with laws such as the Health Insurance Portability and Accountability Act (HIPAA) and enhances the model's generalizability (*Agrawal et al., 2022*; *Supriya & Gadekallu, 2023a*).

Image classification and detection have improved to detect DR, with CNNs playing a crucial role. Privacy protection is crucial, and FL is a decentralized approach. This study investigates three models using federated averaging, federated proximal, and conventional transfer learning frameworks. The models, standard, FedAVG, and FedProx, achieved an accuracy of 92.19%, 90.07%, and 85.81% for DR or non-DR images, respectively (*Attota et al., 2022*; *Nasajpour et al., 2022*). A novel technique for detecting and grading DR in fundus images uses FL, a decentralized method for training DL models without patient data sharing. The method achieves better accuracy, specificity, precision, and F1 score, protecting patient privacy (*Mohan et al., 2023*).

# APPLICATIONS OF ML, DL, AND FL

## Machine learning

ML has several uses in the identification, treatment, and analysis of DR. Automatic DR detection, telemedicine and screening, image segmentation, treatment planning, risk assessment, and progression prediction integration of data and insights, real-time advice and assistance. By analyzing small changes in retinal images, ML models aid in the early detection of DR. By taking into details from scanning and patient history. They forecast the condition's risk and severity. Additionally, ML develops customized treatments by analyzing patient data and suggesting relevant medications. It can segment retinal images and detect anomalies and particular areas of relevance (*Weissler et al., 2021*). Patients in remote locations can benefit from ML models that identify lesions, predict progression, detect DR signs in retinal images, integrate with electronic health records, enable telemedicine, and reduce false positives (*Subramanian et al., 2022*). Furthermore, ML makes it possible to test for DR remotely, facilitating prompt assessment and diagnosis without the need for physical attendance.

## Deep learning

DL techniques enhance traditional AI by enabling computers to learn from data to create intelligent applications. They have been implemented in soft computing and natural language processing. Multi-layer neural networks have increased predictive power in clinical

domains. Deep architectures combine diverse data sets and enable more generalization due to their hierarchical learning structure, prioritizing representation learning over classification accuracy (*Sarker, 2021*; *Kapoor & Arora, 2022*). The development of a distributed patient representation framework has the potential to revolutionize predictive healthcare systems. This system can scale to include millions to billions of patient records and use a single, distributed patient representation to support clinicians' daily activities. The DL framework would be integrated into a hospital's EHR system or other healthcare platforms, and models would be regularly updated to account for changes in study design and clinical trial differences (*Mushtaq & Siddiqui, 2021*; *Alyoubi, Shalash & Abulkhair, 2020*). DL models, which can segment retinal lesions, prevent overfitting, and enhance diagnostic accuracy, are being used to improve DR detection and classification. These models are useful for large-scale screens since they can predict the probability of DR development, further advancing personalized treatment (*Mukherjee & Sengupta, 2023*).

### Federated learning

Decentralized ML techniques such as FL show promise for the diagnosis of DR, particularly when it comes to protecting patient privacy and using data from many sources. Models can be trained on locally maintained data using FL, a technique for DR detection and classification, without exchanging sensitive patient data (*Rauniyar et al., 2023*). It complies with regulations regarding privacy and ethical standards in the healthcare industry while improving performance, lowering transmission bandwidth, and enabling real-time inference (*Li et al., 2021a*; *Supriya & Gadekallu, 2023b*). Decentralized ML techniques such as FL are applied to diagnosing DR, a disease frequently involving sensitive retinal images. This technique allows healthcare organizations to collaborate and train a common model without exchanging patient data (*Li et al., 2021b*). Instead, patient-specific data is preserved by sharing and aggregating just gradients. This method complies with HIPAA and other data privacy laws while improving the model's resilience and generalizability (*Khan et al., 2021*). FL ensures that the model is always learning from fresh data by enabling frequent updates and enhancements without requiring a lot of data exchange (*Bharati et al., 2022*).

## EVALUATION PERFORMANCE METRICS

Evaluation metrics are crucial in developing the best classifiers for data classification. They are used in training and testing stages to improve categorization and choose optimal solutions. During testing, evaluation metrics are used to evaluate classifier performance using confidential data (*Alzubaidi et al., 2021*; *Khan et al., 2020*; *Mateen et al., 2020*). Many of the most widely used evaluation metrics are included in Table 3.

## CHALLENGES AND FUTURE DIRECTIONS

This article emphasizes the challenges in applying new research studies to detect DR using fundus images. The following are some of the challenges and future directions: Resolution of the image, variations in lesion appearance, integration of telemedicine, and incomplete data.

**Table 3 Various metric methods for performance evaluation in DR.**

| Evaluation metrics | Formulation | Definition | References |
|---|---|---|---|
| Accuracy | $\frac{TP+TN}{TP+TN+FP+FN}$ | The proportion of correctly identified objects and non-objects in relation to the total number of objects serves as a measure of accuracy. | *Uppamma & Bhattacharya (2023), Wasekar & Bathla (2021), Islam et al. (2020), Aziz, Charoenlarpnopparut & Mahapakulchai (2023), Tajudin et al. (2022), Vij & Arora (2023).* |
| Sensitivity | $\frac{TP}{TP+FN}$ | Sensitivity is the number of items accurately detected. | *Uppamma & Bhattacharya (2023), Wasekar & Bathla (2021), Islam et al. (2020), Aziz, Charoenlarpnopparut & Mahapakulchai (2023), Tajudin et al. (2022), Vij & Arora (2023).* |
| Specificity | $\frac{TN}{FP+TN}$ | The specificity is determined by the number of non-object classes accurately determined. | *Uppamma & Bhattacharya (2023), Wasekar & Bathla (2021), Islam et al. (2020), Aziz, Charoenlarpnopparut & Mahapakulchai (2023), Tajudin et al. (2022), Vij & Arora (2023).* |
| Precision | $\frac{TP}{TP+FP}$ | The accuracy is found by dividing the number of properly assigned Positive samples by the total number of assigned Positive samples. | *Aziz, Charoenlarpnopparut & Mahapakulchai (2023), Inamullah et al. (2023), Vij & Arora (2023).* |
| Positive Predictive Rate | $\frac{TP}{TP+FP}$ | It is the percentage of positive test results that were appropriately diagnosed. | *Vij & Arora (2023), Alyoubi, Shalash & Abulkhair (2020).* |
| Negative Predictive Rate | $\frac{TN}{TN+FN}$ | The ratio of individuals with a negative diagnosis to those with a negative test result | *Alyoubi, Shalash & Abulkhair (2020).* |
| False positive rate | $1-Spec$ | A false positive result would incorrectly diagnose that a patient has a disease | *Vij & Arora (2023).* |
| F1-score | $\frac{2*TP}{2*TP+FP+FN}$ | It shows how recall and accuracy are connected | *Wasekar & Bathla (2021); Vij & Arora (2023).* |
| AUC | $\int_{-\infty}^{\infty}(T)FPR(T)dt$ | Finding the equation and the operating curve, indicating how sensitive the machine is to the output | *Wasekar & Bathla (2021), Tajudin et al. (2022), Vij & Arora (2023).* |
| Kappa score | $\frac{Acc-Accprob}{1-Accprob}$ | Measuring inter-rater reliability | *Wasekar & Bathla (2021), Vij & Arora (2023).* |

**The investigation of ensemble approaches**: Future research might look into the usefulness of ensemble approaches in DR detection, even though our work concentrated on individual ML and DL models. Multiple models can be combined using ensemble approaches like bagging, boosting, or stacking to increase classification performance, robustness, and generalization.

**The fusion of transfer learning**: For DR detection, transfer learning, a method where previously trained models are improved on a target dataset, has promise. Future research can investigate the use of pre-trained models, including those learned on huge picture datasets like ImageNet, and adapt them to the particular goal of DR detection. By using previously acquired features, this approach can result in less training data being required.

**Incorporation of multi-modal data**: Even though our work focused on fundus images, alternative techniques like optical coherence tomography (OCT) or retinal field data can aid in DR identification. Integrating multi-modal data will help diagnose patients more accurately, enable early DR diagnosis, and give a more thorough evaluation of the condition.

**Addressing interpretability and explainability**: In medical applications, the interpretability and explainability of ML and DL models are crucial. The development of methods to incorporate into the models' decision-making process should be the main emphasis of future studies. This will facilitate more effective collaboration between human experts and automated systems by assisting physicians and other healthcare professionals

in getting to know the forecasts. We also consider how new technologies like explainable artificial intelligence (XAI) and reinforcement learning advance the field of DR detection.

## CONCLUSION

The review of ophthalmology studies focuses on publicly available datasets, classification methodologies and applications, and performance measures in DR. It identifies research challenges and future directions, such as the lack of focus on the location of various DR lesions in existing DL models for DR severity diagnosis. Future research could expand to include transfer learning, explainable AI, multi-task learning, and domain adaptation for early detection of DR. Intelligent health monitoring technologies can reduce diagnosis time and cost, allowing faster patient communication. The authors believe this review would benefit scientists and medical practitioners working in DR detection.

### Funding
The authors received no funding for this work.

### Competing Interests
The authors declare there are no competing interests.

### Author Contributions
- Dasari Bhulakshmi conceived and designed the experiments, performed the experiments, analyzed the data, performed the computation work, prepared figures and/or tables, authored or reviewed drafts of the article, and approved the final draft.
- Dharmendra Singh Rajput conceived and designed the experiments, performed the experiments, analyzed the data, performed the computation work, prepared figures and/or tables, authored or reviewed drafts of the article, and approved the final draft.

### Data Availability
 This is a literature review.

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
