# Peer review of "A systematic review on diabetic retinopathy detection and classification based on deep learning techniques using fundus images"

_PeerJ Computer Science, doi:10.7717/peerj-cs.1947_

## Round 0.1 · original submission · Major Revisions

The reviewers have substantial concerns about this manuscript. The authors should provide point-to-point responses to address all the concerns and provide a revised manuscript with the revised parts being marked in different color.

**Language Note:** The review process has identified that the English language must be improved. PeerJ can provide language editing services - please contact us at copyediting@peerj.com for pricing (be sure to provide your manuscript number and title). Alternatively, you should make your own arrangements to improve the language quality and provide details in your response letter. – PeerJ Staff

Reviewer 1 ·

Basic reporting

The authors summarized deep learning techniques that have been applied in diabetic retinopathy detection and classification. The authors introduced the objective and motivation clearly. They summarized the datasets that have been used in various research. They also provided the evaluation methods that are widely used in deep learning techniques. The review is comprehensive. However, there are a lot of grammatical mistakes, please check the sentence clarity to improve the manuscript.
1. Please check the grammar and make sure punctuation is used correctly. Please also capitalize the first letter of the first word whenever you begin a sentence. For example, please capitalize O of "outline" in the abstract. Another example is “In Fig. 1, discuss the status of diabetes, present and future, through this diabetes prediction chart” (line 29), please add “we” before “discuss”.
2. Please add “million” in figure 1. This figure is misleading since the authors didn’t specify the unit.

Experimental design

The survey methodology is comprehensive and unbiased. Sources are adequately cited. The sections are well organized. However, there are some redundant sentences in each section. For example, the authors introduced DR in every section which is not necessary. Please remove unnecessary sentences to make the manuscript easier to read.

Validity of the findings

The authors summarized datasets and some implementations of deep learning techniques. The authors also described current challenges and future directions. It would be great if the authors could briefly summarize the pros and cons of different deep learning techniques in a table.

Reviewer 2 ·

Basic reporting

This paper is a systematic review on diabetic retinopathy detection and classification based on deep learning techniques using fundus images. The review aims to examine the advancements in automated scientific methods for detecting and classifying diabetic retinopathy, including both conventional and deep learning approaches. It discusses various neural network designs, such as machine learning, recurrent neural networks, generative adversarial networks, and federated learning, as well as convolutional neural networks and their variations. The review also outlines potential research methods, challenges, and future directions in the field. The document provides instructions for structuring the review and offers tips for effective reviewing. This topic is significant, yet it is not full of details about the deep learning techniques comparison and relevant analysis for the techniques advance in diabetic retinopathy detection and classification. The general quality of this paper is not solid and robust in models discussion. It is a very general introduction for this topic. The systematic review need more details and comprehensive comparison in difference techniques and the comments on the tendency of the tendency of techniques development in this area.

Experimental design

As above

Validity of the findings

As above

Additional comments

As above

Reviewer 3 ·

Basic reporting

The authors tried to provide a systematic review on using deep learning (DL) based machine learning (ML) methods for diabetic retinopathy (DR) detection. In the context of DR detection using ML methods, the authors failed to:
1). Explain what the existing methods are for DR detection, if no ML methods involved (basically how doctors detect DR in the fundus images?).
2). Explain what signals doctors or ML methods should specifically look for in the retina fundus images (healthy retina vs unhealthy retina contrast, using detailed image/data examples).
3). Following 1) and 2), Explain why deep-learning can play a role here for the DR detection in the fundus images.
Instead, authors spent too much effort on explaining basics concepts of diabetes (instead of more details on the DR directly with real data samples), very shallow and basic introduction of existing deep learning methods, instead of detailed application examples of those DL methods in the DR detections and systematic summary of those specific examples(like what exact model/neural network structures were used in the field of DR detection and specific performance metrics such as AUCROC/Accuracy/Sensitivity of those DL models). And such shallow introduction on diabetes and/or deep learning methods is almost repeated in every section and subsection (examples: line 89, line 110-112, line 168, line 192-193, all repeating DR is a result of diabetes).
Additionally, the field of ML/DL assisted DR detection has been extensively and thoroughly reviewed recently (ref 1: https://www.ncbi.nlm.nih.gov/pmc/articles/PMC9914068/, ref 2: https://www.nature.com/articles/s41467-021-23458-5). Both of the reviews mentioned above provided a more professional systematic review with detailed examples (such as model structures, model performance, specific detection examples) than the current article.

Experimental design

The references management of this review article is in chaos. The first 6 references (line 301-306) of this article do not follow reference standard at all: with no url links pointing to the public accessible online resources, repeated references and wrong reference indexes. More concerningly, the whole reference citing structure of this article do not follow academia standards: the reference starts not with 1, but with 47 in a discontinuous fashion (line 29) and such chaotic citing pattern (discontinuous ref index naming, content mismatch with references) is ubiquitous throughout the whole article (just to name a few: line 29, 33, 37, 49, 97, 195 etc.).
As for the Survey Methodology section, the authors
1). Failed to list detailed bibliographic databases sources with link;
2). Failed to describe detailed inclusion/exclusion criteria, like literature time scope of interest, detailed title and abstract matching criteria, not just briefly mention “Then, we screen all the retrieved papers based on their titles. We have excluded the papers that did not coincide with the scope of the present context.” Line 83-84. The authors need to give detailed exclusion criteria not only based on the title but also on the abstract.
3. It is highly questionable that authors just used “ML in DR”, “DL in DR”, “FL in DR” (line 82) with full abbreviations to retrieve relevant literatures on the deep learning (DL) based machine learning (ML) methods for diabetic retinopathy (DR) detection. As I briefly searched on Google, none of the queries above return DR relevant information on the top results.
Additionally, in the section and sub-section structure of the whole review, authors tend to place ML in the parallel position of Deep-Learning (DL), such as line 191, line 224/236. However, in general, as author indicated in the Figure 2, DL is just subsection of ML. The current subsection structure of the whole review is out of order.

Validity of the findings

The review article is lack of supportive arguments to meet the goals set out in the Introduction.
1). The authors failed to deep-dive into details of each model’s application in the DR detection, such as data examples, model structures and model performance. Authors stopped at very shallow and basic introduction of each DL model category (Section “Types of Deep Learning”, section “FEDERATED LEARNING IN DIABETIC RETINOPATHY”, section “APPLICATIONS OF DL, ML AND FL”, section “EVALUATION PERFORMANCE METRICS”), which is far below standard for a domain review paper.
2). There are 4 figures in the article. However, none of the figures are directly related to the DL based DR detection, which adds minimum value to the whole review article. Figure 1 is about diabetes patients evolve as a function of time, which has wrong units. Figure 2 is shallow introduction of relationship between AI,ML and DL, has no direct relation to the the DR topic. Figure 3 is a very basic comparison of DL vs ML, has minimum meaning to the whole paper. Figure 4 is another shallow example of so call CAD system, with no relation to the DR at all.
3). The 3 tables in the article provides minimum summary information. Table 1 failed to give detailed model structure of each DL applications in the DR detection and failed to give detailed performance metrics for each model. Instead of giving detailed model performance in each DR detection application, Table 3 gave a very brief and basic introduction to the binary classification ML task evaluation metric definition, which is completely not necessary.
4). The article needs extensive language edits for grammar errors. For example (just to name a few),
a. in the abstract, line 21, missing subject (“outline the potential research method…”);
b. line 22, missing subject (“Discusses the challenges…”) and wrong use of “,” at the end of the sentence
c. line 31, wrong tense use of “will suffer from diabetes”
d. line 67. Tense conflicts, “section represents, section 3 related”
In general, the language used in the article falls below the bar for an academic review paper.

---

## Round 0.2 · Minor Revisions

Most of concerns have been addressed. There are only some remaining minor concerns that need to be addressed.

Reviewer 2 ·

Basic reporting

I have no further comments on this paper.

Experimental design

No

Validity of the findings

No

Additional comments

No

Reviewer 4 ·

Basic reporting

The manuscript entitled "A systematic review on diabetic retinopathy detection and classification based on deep learning techniques using fundus images" requires minor revisions before it can be accepted for publication.

More biological and medical understanding of diabetic retinopathy and more information on the quality control of images related to this disease should be provided. This will provide more context and support for the arguments presented in the manuscript.

Experimental design

it would be helpful if the authors could cover more deep learning innovations, including graph neural networks and the impacts and outlook of large language models like ChatGPT. This will help readers understand the state-of-the-art techniques in the field and their potential applications in diabetic retinopathy detection and classification.

Validity of the findings

The authors of the article have conducted a validity survey on diabetic retinopathy detection and classification based on deep learning techniques using fundus images. This is an important area of research as diabetic retinopathy is a leading cause of blindness and early detection can help prevent vision loss. By using deep learning techniques and fundus images, the authors aim to improve the accuracy and efficiency of diabetic retinopathy detection and classification.

Additional comments

no comment

---

## Round 0.3 · accepted · Accept

The authors have addressed all concerns and I recommend accepting this manuscript.